# Does Early Diagnosis and Treatment Alter the Clinical Course of Wolman Disease? Divergent Trajectories in Two Siblings and a Consideration for Newborn Screening

**DOI:** 10.3390/ijns11010017

**Published:** 2025-02-25

**Authors:** Maria Jose de Castro Lopez, Fiona J. White, Victoria Holmes, Jane Roberts, Teresa H. Y. Wu, James A. Cooper, Heather J. Church, Gemma Petts, Robert F. Wynn, Simon A. Jones, Arunabha Ghosh

**Affiliations:** 1Manchester Centre for Genomic Medicine, Manchester University NHS Foundation Trust, Oxford Road, Manchester M13 9WL, UK; maria.lopez@mft.nhs.uk (M.J.d.C.L.); fionajwhite45@gmail.com (F.J.W.); victoria.holmes@mft.nhs.uk (V.H.); jane.roberts@ndorms.ox.ac.uk (J.R.); hoiyee.wu@mft.nhs.uk (T.H.Y.W.); heather.church@mft.nhs.uk (H.J.C.); arunabha.ghosh@mft.nhs.uk (A.G.); 2Department of Therapy and Dietetics, Manchester University NHS Foundation Trust, Oxford Road, Manchester M13 9WL, UK; 3Department of Paediatric Histopathology, Royal Manchester Children’s Hospital, Oxford Road, Manchester M13 9WL, UK; gemma.petts@mft.nhs.uk; 4Department of Blood and Marrow Transplantation, Royal Manchester Children’s Hospital, Oxford Road, Manchester M13 9WL, UK; robert.wynn@mft.nhs.uk

**Keywords:** Wolman disease, enzyme replacement therapy, dietary substrate reduction, early diagnosis, early treatment, newborn screening

## Abstract

Wolman disease (WD) is a lethal disorder defined by the deficiency of the lysosomal acid lipase enzyme. Patients present with intestinal failure, malnutrition, and hepatosplenomegaly. Enzyme replacement therapy (ERT) with dietary substrate reduction (DSR) significantly improves survival. We sought to determine the outcomes of two siblings with WD treated after the onset of symptoms (sibling 1) and presymptomatic (sibling 2). A chart review was conducted on two siblings with WD treated with ERT and DSR at 4 months of age (sibling 1) and immediately after birth (sibling 2) to determine clinical outcomes based on survival, laboratory results, growth, dietary records, and gut biopsies. Sibling 1 presented with hepatosplenomegaly and liver dysfunction and developed hemophagocytic lymphohistiocytosis despite treatment. She received a bone marrow transplant at 8 months of age but died at 13 months. Sibling 2 is alive at 16 months of age with height, weight, and MUAC above the 95th centile, fully orally fed, with no gastrointestinal symptoms, normal liver function, and normal oxysterols. Sibling 2 duodenal biopsies show normal villus architecture with no foamy macrophage infiltration. Initiation of treatment prior to the onset of symptoms can prevent clinical manifestations and increase survival. The divergent trajectory in these siblings raises the question of WD’s candidacy for newborn screening.

## 1. Introduction

Wolman disease (WD) is an ultrarare (1 in 350,000 to 1 in 500,000) autosomal recessive disorder caused by the deficiency of lysosomal acid lipase (LAL) enzyme, which leads to the accumulation of cholesteryl esters and triglycerides [1,2]. WD is characterized by early-onset growth failure, vomiting, diarrhea, hepatosplenomegaly, and rapidly progressive liver failure [3,4]. Another common initial presentation includes a secondary hemophagocytic syndrome [5,6,7]. When untreated, WD leads to death within the first 6 months of life [3]. Diagnosis is based on clinical symptoms, measurement of LAL enzyme activity using dried blood spots (DBS) or leukocytes, and molecular genetic confirmation by sequencing of *LIPA* [1,2].

Sebelipase alfa (Kanuma^®^, Kanuma™) has been approved since 2015 in several countries as a long-term enzyme replacement therapy (ERT) for patients diagnosed with Wolman disease. Dietary substrate reduction, with all dietary lipids minimized to as close to zero as possible, is a vital part of treatment.

Due to the nonspecific initial symptoms and the ultrarare nature of WD, the diagnosis is frequently delayed. Frequent misdiagnoses include cow’s milk protein intolerance, sepsis, and HLH [3,4,5,6,7]. There are some case reports in the literature presenting patients with Wolman disease primarily diagnosed as HLH, a life-threatening hyperinflammatory disorder. Primary HLH is basically treated by immune suppression, which is provided by steroids and chemotherapy, but treatment for secondary HLH related to WD should primarily trigger the underlying condition, which means starting ERT as soon as possible. While delays in the diagnosis of ultrarare disorders are often measured in years, even 1 to 2 weeks of delay in treatment initiation can be fatal in WD [8,9,10]. Correlating ERT timing to the effect on the clinical course and progression of symptoms can be challenging. Recently, a French cohort of 5 patients with Wolman Disease treated before 3 months of age reported a 100% survival [10], which contrasts with the survival rate of VITAL and CL08 clinical trials. The median age of the first dose in the clinical trials was 3 and 2.8 months, respectively, with a range of 1 to 6 and 0.5 to 4 months [8,9]. These good results can be explained by the high proportion of positive family history (3/5) in the French cohort, leading to early diagnosis and better clinical status at ERT and DSR initiation. As with many lysosomal storage diseases, initiation of treatment prior to the onset of irreversible symptoms may be advantageous. Moreover, the present ability to accurately measure LAL enzyme level from a blood spot using a specific substrate and ultraperformance liquid chromatography-mass spectrometry potentially further qualifies this disease as a future candidate for newborn screening [11,12,13]. However, no country currently includes WD in their newborn screening (NBS) programs, and a novel algorithm for assessing disorders for inclusion in programs based on the availability of therapy was proposed in 2022 [14].

We present the trajectory of two siblings with Wolman disease treated at 4 months of age (sibling 1) and immediately after birth (sibling 2) with DSR from birth and ERT started in the first week of life. Assessing treatment outcomes in siblings with similar genetic backgrounds and expected rates of disease progression unravels the full potential benefits of sebelipase alfa, as has been described previously for other ERT and lysosomal disorders [15,16,17]. Significant divergent trajectories when ERT with DSR is initiated before the onset of symptoms would support the candidacy of Wolman disease for newborn screening.

### Royal Manchester Children’s Hospital (RMCH)

RMCH is a specialist center for the diagnosis and treatment of inherited metabolic disorders, including the delivery of HCT. Since 2012, 13 patients with Wolman disease have been treated in Manchester; of these, five individuals have died, and eight are alive. RMCH participated in both VITAL (Safety, Tolerability, Efficacy, Pharmacokinetics, and Pharmacodynamics of Sebelipase Alfa in Children with Growth Failure Due to Lysosomal Acid Lipase Deficiency, NCT01371825) and CL08 (Clinical Study In Infants With Rapidly Progressive Lysosomal Acid Lipase Deficiency, NCT02193867) trials. All patients with Wolman disease at RMCH are under the care of a multidisciplinary team, including a specialist metabolic dietitian. At diagnosis, dietary substrate reduction (DSR) via minimal lipid intake (aim < 1 g/day) is initiated alongside ERT [18]. In clinically presenting cases, this is most often initially as modified total parenteral nutrition, transitioning over several months to amino acid based minimal fat enteral feeds, usually as continuous tube feeds. Some patients with Wolman disease may benefit from bone marrow transplant (BMT) [19,20].

## 2. Materials and Methods

Written consent for the publication of patient data was obtained from the family of both siblings. Demographic information and clinical data were collected from medical records, and hematological and biochemical data were obtained from hospital laboratory records.

### 2.1. LAL Enzyme Activity

LAL activity was measured in mixed leucocytes in both siblings using the artificial substrate 4-methylumbelliferyl-palmitate (NBS Biologicals, Huntingdon, UK) based on the previously published protocol [21,22].

### 2.2. Molecular Genetic Diagnosis

Molecular genetic analysis was performed by Sanger sequencing. All exons and intron-boundaries, as well as 3′ and 5′-UTR regions for the LIPA gene, were amplified by homemade design PCR. Products were sequenced by capillary electrophoresis using a SeqStudio analyzer (Applied Biosystems, Waltham, MA, USA), and results were compared with the GeneBank database’s reference sequence (NG_008194.1).

### 2.3. Plasma Oxysterols Levels

Serial measurements of plasma oxysterol concentrations were used as a surrogate disease biomarker. Cholestane-3ß,5α,6ß-triol was quantified by liquid chromatography and coupled tandem mass spectrometry as previously reported [23,24].

## 3. Results

### 3.1. Siblings

These two sisters are both homozygous for a missense LIPA variant c.824C>T p.(Ser275Phe) and were markedly deficient in LAL activity in leukocytes [31 nmol/mg/h and 35 nmol/mg/h for sibling 1 and sibling 2, respectively (reference range 350–2000 nmol/mg/h)]. Sibling 1 was diagnosed with Wolman disease at the age of 4 months, at which time she had classical features of the disease. Her previous diagnosis led to the diagnosis in the younger sister shortly after birth. (Figure 1A–D).

### 3.2. Sibling 1

Had a history of growth failure, massive hepatosplenomegaly, coagulopathy, thrombocytopenia, and anemia. She was commenced on ERT and DSR at 4 months of age. ERT dosing was according to the LAL CL08 protocol, starting at 1 mg/kg weekly and increasing after two weeks to 3 mg/kg weekly, and after a further four weeks to a maximum of 5 mg/kg weekly [8]. Plasma oxysterols were markedly elevated at baseline (338–429 ng/mL, reference range 9.6–37 ng/mL) and were still substantially elevated after 2 months of treatment (209 ng/mL). Liver function tests were abnormal at baseline (ALT 124–192 U/L) and deteriorated over the first 2 months of treatment (maximum ALT 466 U/L). Clinically, she manifested a continuing hyperinflammatory picture with elevated inflammatory markers and multiple episodes of pyrexia without evidence of bacterial infection. At 6 months of age, she was treated under the HLH 94 protocol (etoposide, dexamethasone, ciclosporin). Her clinical picture partly improved, and she underwent a bone marrow transplant [20]. Despite engraftment, she continued to have evidence of an ongoing inflammatory process, with splenomegaly, persistent fever, cytopenia, hyperferritinemia, and elevated cytokines (sCD25, IL-1ß). Organomegaly and gut function did not improve. There was no evidence of an antibody response to sebelipase. She died at the age of 15 months after developing multiorgan failure in the context of gram-negative sepsis. Sibling 1 underwent upper and lower GI endoscopy at 9 months of age due to an episode of gastrointestinal bleeding. Histology from biopsies taken during this procedure demonstrated no cause for the bleeding. The duodenal biopsies were abnormal and showed the expansion of the duodenal lamina propria by vacuolated macrophages with focal villous blunting (Figure 2A,B). Despite multimodal treatment and being gastrostomy fed, Sibling 1 MUAC was always <11.5 cm. WHO defines a MUAC <11.5 cm as severe acute malnutrition with a high degree of morbidity.

### 3.3. Sibling 2

This patient had diagnostic testing at birth due to the known family history of her sibling. Antenatal ultrasound at 32/40 gestation demonstrated enlarged adrenal glands and a mildly echogenic bowel with one dilated bowel loop. She was started from birth on DSR with a minimal-fat (<1%) infant formula (Low Fat Module, Nutricia, Wiltshire, UK) and long-chain polyunsaturated fatty acid supplement (Key OmegaTM, Vitaflo, UK), and she received her first dose of sebelipase at 4 days of age. Baseline plasma oxysterols were normal (15.6 ng/mL) and remained within or below the reference range throughout follow-up. Liver transaminases were normal at baseline and throughout follow-up. Ferritin was mildly raised at baseline (306 µg/L, reference range 12–150) but normalized after 2 months of therapy. At the latest follow-up (age 16 months), she is thriving with length and weight on the 96th centile for age, with mid-upper arm circumference at 1 S.D. above the mean. She has normal motor development. There are no gastrointestinal symptoms, and she is fully orally fed and has started weaning on to a minimal-fat (aim < 5 g/day) diet. Sibling 2 underwent upper GI endoscopy at 1 year of age as per local clinical practice for monitoring outcomes and treatment response. Biopsies from D2 demonstrated essentially normal duodenal mucosa with no evidence of a significant foamy macrophage infiltrate and normal villous architecture (Figure 2C,D).

## 4. Discussion

WD is a rare, progressive, infantile onset disease caused by insufficient LAL activity and leading to death with a median age of 3.5 months [2]. The disease hallmarks include liver failure, hepatosplenomegaly, steatorrhea and malabsorption, and adrenal calcification due to extensive storage of cholesteryl esters and triglycerides in the lysosomes of Kupffer cells, hepatocytes, and macrophages [3]. Sebelipase alfa is licensed for the long-term treatment of infants with WD after receiving marketing authorization from the Food and Drug Administration and the European Medicine Agency in 2015, following an accelerated assessment due to unmet medical needs.

The advent of commercially available enzyme replacement therapy for WD has led to dramatic improvements in survival. Unlike some lysosomal disorders involving bone or brain disease, many of the features of WD seem to be reversible with ERT. Established inflammation, however, can be refractory to treatment with ERT alone. While the survival benefit of sebelipase alfa was clear from the two clinical trials, even in the second trial, some infants died [9]. This was recognized to be often those infants presenting in later stages of the disease, with significant inflammation and more severe end organ damage. Sadly, not all treated children survive, and despite improved outcomes, they will need ongoing therapy with DSR and additional immune modulation or transplant [19,20] in those not meeting therapeutic goals.

WD is a medical emergency with rapid disease progression and early death. Delayed diagnosis is common, as this condition is underrecognized as a cause of marked growth failure and rapidly progressive liver failure in infants and currently is not included in newborn screening. Accordingly, many patients commence treatment when they already have a considerable disease burden, leading to significant morbidity and mortality despite ERT [6,7,8] and DSR. These siblings demonstrate the benefits of early diagnosis in generating an improved outcome. Demaret et al. [10] showed that early ERT initiation (before 5 months of life) with DSR on five patients with Wolman disease allowed 100% survival. In the era of expanding newborn screening, it is equally important to consider whether this rare but life-threatening disease may benefit from rapid and earlier diagnosis. Moreover, continued treatment with DSR from birth has been associated with tolerance of intact proteins (as opposed to most clinically presenting cases requiring amino acid based feeds), normal growth, absence of gastrointestinal symptoms, and negated the need for assisted tube feeding, required for a long term in many clinically presenting cases. With regards to the lived experiences of parents whose child had a diagnosis of LALD, Hassall et al. [25] highlighted the significant impact that a diagnosis of LALD has on parents, uncovering their psychological experiences from the point of diagnosis to living with a high degree of uncertainty. The study explored how parents have been able to come to terms with the diagnosis, facilitated by their child being able to live an unrestricted life as a consequence of treatment, giving families hope.

NBS is a public health measure for a group of diseases as defined on a country-to-country basis. The criteria for assessing which diseases to include are generally accepted to be those defined by Wilson and Jungner [26], and WD fulfills these criteria. Evidence of early therapy versus late onset therapy is one of the key principles of NBS and must be established for each disease considered. One of the most important W&J criteria in this respect is probably ‘There should be an accepted treatment for patients with recognized disease’. The aspect of an accepted treatment or treatability consists of three main requisites: (1) the presence of treatment for this disorder, (2) the approval of this treatment by the FDA/EMA, and (3) financial coverage or reimbursement by standard healthcare. Without treatability, no disorder will be approved to be selected for NBS.

From a technical NBS perspective, a new high-throughput functional assay of LAL by ultra-high pressure liquid chromatography-tandem mass spectrometry (UPLC-MS/MS) is currently under evaluation [27]. This assay deploys a highly specific LAL substrate and requires one 3.1 mm diameter bloodspot disc with one incubation step. It is faster than the alternative bloodspot LAL method using a nonspecific fluorometric substrate for lipase, where LAL is calculated by subtracting lipase activities incubated with and without lalistat 2, a specific LAL inhibitor [28]. A multiplexed UPLC-MS/MS method that screens for a panel of treatable lysosomal storage disorders, including LAL deficiency, is currently under evaluation in a prospective newborn screening study in New York, US [29].

Setting a highly sensitive screening cutoff level for a screening test evaluation is usually dependent on the availability of age-matched positive newborn bloodspots for the condition screened for. Finding positive newborn bloodspots can be a real challenge for ultrarare conditions such as WD when the proposed first-tier NBS test is to measure enzyme activity. The short-term and long-term stability of the enzyme in the bloodspot should be taken into consideration when evaluating the screening test. It has been shown that the LAL enzyme was mostly preserved at −20 °C, with approximately a 10% drop in activity after 100 days; a more rapid loss of 10% in 7 days was observed when the bloodspot was stored at room temperature [29]. It is possible to identify affected cases in bloodspots stored at room temperature after a few weeks in a routine diagnostic setting as there is a big difference between the affected and the normal reference ranges (affected <0.03 and normal range 0.37–2.3 nmol/punch/h, personal communication from Dr Marianne Barr, Glasgow UK). However, the long- term stability issues limits the availability of good quality, positive bloodspot for WD NBS test validation.

A further consideration is the assessment of the test to differentiate the severe and more attenuated clinical phenotypes. A collaborative, multicenter approach would enable the gathering of real-world evidence to satisfy the very strict criteria defined by the newborn screening program centers. The same approach has already shown some success in escalating the formal review of MLD for newborn screening in several countries, such as the US, Netherlands, Italy, and the UK [30,31,32].

The comparative observations after 16 months of ERT in this unique sibling pair provide convincing evidence of substantial differences in the impact of sebelipase and DSR in WD when therapy is initiated prior to disease manifestations. Notably, impaired growth, gastrointestinal symptoms, and inflammation continued to progress despite ERT, DSR, and HSCT in the sibling who started therapy at 4 months of age, whilst the younger sibling, who started therapy presymptomatically in the first week of life, shows height, weight, and MUAC above the 95th centile, no gastrointestinal symptoms, normal liver function, and normal oxysterols. Comparison of the sibling’s duodenal biopsies echoes this clinical difference with the more severely affected Sibling 1 demonstrating a marked foamy lipid laden macrophage lamina propria infiltrate, histologically typical of this disease, compared to relatively normal duodenal histology in Sibling 2. While discordant clinical courses can be seen even in siblings with the same monogenic disorder, this is usually subtle and less evident with such severe phenotypes.

The other difference in the treatment of these two siblings was in the doses of sebelipase used. It is possible that by starting on 5 mg/kg in sibling once or even twice weekly sebelipase as recently described [33], we could also have saved sibling 1; this, however, was not possible due to the constraints of the trial protocol and knowledge at the time. We acknowledge the limitation that this report is only based on two individuals who were reported retrospectively. However, due to the extreme rarity of this disease, it is recognized that even these limited data can be highly instructive.

## 5. Conclusions

This case report illustrates that early treatment substantially modifies the natural history of WD and brings to the forefront the importance and potential impact that neonatal screening and early treatment initiation could have on this rapidly progressive and still life-threatening disease.

## Figures and Tables

**Figure 1 IJNS-11-00017-f001:**
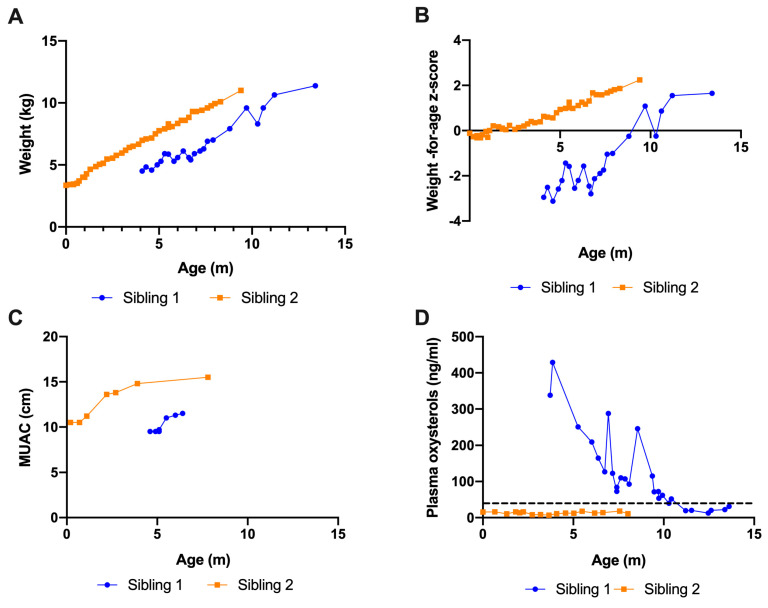
(**A**) Weight gain in both siblings, (**B**) weight gain for age Z score, (**C**) MUACs progression, and (**D**) plasma oxysterols levels in both siblings.

**Figure 2 IJNS-11-00017-f002:**
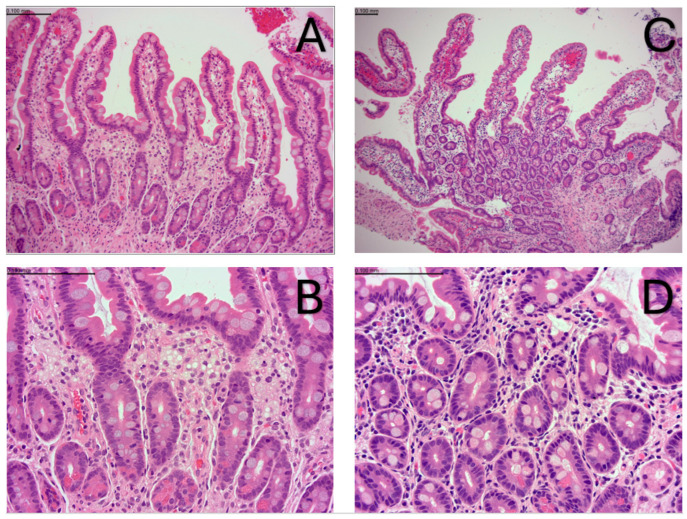
All haematoxylin and eosin stained FFPE (routine processing). Microscope Leica DM4B with Tucsen Mosaic2.2 camera system. (**A**,**C**) magnification ×100. (**B**,**D**) magnification ×400. Scale bars included in images. (**A**,**B**) Sibling 1 at 9 months of age, duodenal biopsy. (**A**) Duodenal mucosa showing expansion of the lamina propria with foamy macrophages and associated loss of normal lamina propria cellularity. (**B**) Higher power view of lamina propria foamy macrophages. (**C**,**D**) Sibling 2 at 1 year of age, duodenal biopsy. (**C**) Normal duodenal mucosa. (**D**) Higher power view of lamina propria showing normal cellularity and lack of foamy macrophages.

## Data Availability

Data are available on request.

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
