# Peer review of "Does Early Diagnosis and Treatment Alter the Clinical Course of Wolman Disease? Divergent Trajectories in Two Siblings and a Consideration for Newborn Screening"

_2409-515X, 2025, doi:10.3390/ijns11010017_

Round 1
Reviewer 1 Report
Comments and Suggestions for Authors
The manuscript is well-written, addressing an important topic in neonatal screening and treatment for Wolman disease (WD). The study offers valuable insights into the benefits of early diagnosis and treatment, supported by a comparison of two sibling cases. The presentation of the two cases effectively illustrates the divergence in clinical outcomes due to differences in treatment initiation.
Several improvements could be made in the manuscript:
- Intro - Could be shortened in general desrciptions of WD but expanded on why WD is underdiagnosed and the current limitations in newborn screening.
-Methods, Results - This study is implying the transformative potential of early diagnosis and treatment for WD. To build on these findings, a systematic review of the literature is recommended to be added on this particular aspect, which is crucial for the aims of the authors, since it is otherwise quite hard to make generalisations basing only on two cases. Such an effort could thus provide a much stronger foundation for integrating WD into newborn screening programs and optimizing management strategies for affected individuals. This way, the manuscript could also be a better fit for this journal.
- Discussion - Explicitly state the limitations of the study, such as its small sample size and the retrospective design.
- I would suggest that the presentation of two cases is condensed somewhat (currently there are many very general aspects and descriptions of WD and even some redundancy in case presentations), and that second part of the manuscript is introduced presenting a ideally systematic review of literature focusing on how early (neonatal) detection is crucial for acceptable outcomes. Case for WD newborn screening should be built more generally.
Author Response
Response to reviewers,
We would like to thank the reviewers for their helpful comments and time taken to review and improve our manuscript. We address their concerns and revise our manuscript accordingly below and in the attached/uploaded revised manuscript.
Reviewer 1
The manuscript is well-written, addressing an important topic in neonatal screening and treatment for Wolman disease (WD). The study offers valuable insights into the benefits of early diagnosis and treatment, supported by a comparison of two sibling cases. The presentation of the two cases effectively illustrates the divergence in clinical outcomes due to differences in treatment initiation.
Several improvements could be made in the manuscript:
- Intro - Could be shortened in general descriptions of WD but expanded on why WD is underdiagnosed and the current limitations in newborn screening.
We have shortened the introduction and been clearer regarding underdiagnosis of WD (we do not think this is under-diagnosed per se but rather late diagnosed) and made more comments on the current state of newborn screening for WD.
-Methods, Results - This study is implying the transformative potential of early diagnosis and treatment for WD. To build on these findings, a systematic review of the literature is recommended to be added on this particular aspect, which is crucial for the aims of the authors, since it is otherwise quite hard to make generalisations basing only on two cases. Such an effort could thus provide a much stronger foundation for integrating WD into newborn screening programs and optimizing management strategies for affected individuals. This way, the manuscript could also be a better fit for this journal.
Given the relatively new availability of therapy for WD and sparcity of the literature we have reviewed all published cases of therapy with Sebelipase. We have predominantly focussed on the case series and drawn out any conclusions regarding early vs late diagnosis and the effects on therapeutic outcomes. We have compared in the introduction the clinical trial data (VITAL and CL08, Vijay et al) to the only subsequent case series (Demaret et al) published from France. The other case series published (Potter et al) focuses on combination therapies rather than diagnosis and many of the patients reported in this were participants in the clinical trials.
- Discussion - Explicitly state the limitations of the study, such as its small sample size and the retrospective design.
We acknowledge these limitations.
- I would suggest that the presentation of two cases is condensed somewhat (currently there are many very general aspects and descriptions of WD and even some redundancy in case presentations), and that second part of the manuscript is introduced presenting a ideally systematic review of literature focusing on how early (neonatal) detection is crucial for acceptable outcomes. Case for WD newborn screening should be built more generally.
We have reduced the information contained in the case presentations and removed any duplication or redundancy.
Evidence for benefit of early therapy versus later onset of therapy is one of the key principles of newborn screening (Wilson and Jungner) and must be established for each disease. This is discussed in the discussion section and the case for including WD is presented more systematically.
Reviewer 2 Report
Comments and Suggestions for Authors
see word document

Author Response
Reviewer 2
The paper describes two siblings affected by LAL deficiency, the first diagnosed at a markedly symptomatic stage with a fatal course despite triple therapy with DSR, ERT and HSCT and the sibling suspected to be affected by fetal ultrasound with postnatal confirmation and immediate treatment initiation with DSR and ERT and good overall health at age 16 months. Authors stress the importance of an early diagnosis and discuss methods and implementation into national screening programms. This paper illustrates the need of an early diagnosis for optimal impact of available treatment.
There are a few points that need attention
The first child underwent triple therapy, including HSCT. Authors should please outline the indication for HSCT- was this done due to persistent HLH or for severity of symptoms?
We have clarified the indication for the HSCT was continuing evidence of hyper inflammatory disease.
I could not find results of LAL activity in either of the siblings.
Added a sentence about the deficient LAL activity in both siblings
Though early diagnosis and treatment clearly benefitted the second sibling, authors may want to outline divergent clinical course is possible in siblings affected by the same autosomal recessive disorder.
Addressed in the text as a potential though not likely confounder .
Authors should outline the indication for duodenal biopsy in sibling 2 and comment if this should be part of regular disease monitoring.
Sentence added clarifying local practice is 1-2 yearly biopsies to monitor treatment outcomes as this is a relatively new therapy and not well understood disease.
For the easyness of reading discussion would benefit from a more clear structure of eg. first treatment options for WD, the change in their strategies over time and still existing limitations.
Aspects on the cases presented and finally address newborn screening.
Discussion structured differently with more details at the end on newborn screening as per both reviewers
Part of the approach to newborn screening for LSDs and current methods to detect LAL deficiency could be transferred to the intro section.
Discussion line 188-191-please insert citation(s).
Reference added
Discussion line 194 typo? …median age of 3.5 months 12
Corrected.
Round 2
Reviewer 1 Report
Comments and Suggestions for Authors
The issues raised were appropriately addressed.